# DATA TRANSFER APPROACHES TO IMPROVE SEQ-TO-SEQ RETROSYNTHESIS

## ABSTRACT

Retrosynthesis is a problem to infer reactant compounds to synthesize a given product compound through chemical reactions. Recent studies on retrosynthesis focus on proposing more sophisticated prediction models, but the dataset to feed the models also plays an essential role in achieving the best generalizing models. Generally, a dataset that is best suited for a specific task tends to be small. In such a case, it is the standard solution to transfer knowledge from a large or clean dataset in the same domain. In this paper, we conduct a systematic and intensive examination of data transfer approaches on end-to-end generative models, in application to retrosynthesis. Experimental results show that typical data transfer methods can improve test prediction scores of an off-the-shelf Transformer baseline model. Especially, the pre-training plus fine-tuning approach boosts the accuracy scores of the baseline, achieving the new state-of-the-art. In addition, we conduct a manual inspection for the erroneous prediction results. The inspection shows that the pre-training plus fine-tuning models can generate chemically appropriate or sensible proposals in almost all cases.

## 1 INTRODUCTION

Retrosynthesis, first identified by Corey & Wipke (1969), is a fundamental chemical problem to infer a set of *reactant* compounds that can be synthesized into a desired *product* compound through a series of chemical reactions. The search space of sets of compounds is innately huge. Further, a product compound can be synthesized through different series of reactions from different reactant compound sets. Such difficulties require the huge efforts of human chemical experts and the large knowledge base to build a retrosynthesis engine for years. Thus, expectations of machine-learning (ML) based retrosynthesis engines is growing in recent years. The need for the retrosynthesis becomes intensive in these days along with the development of *in silico* (computational) chemical compound generations (Jin et al., 2018; Kusner et al., 2017), which are also applied to new drug discovery for COVID-19 (Cantürk et al., 2020; Chenthamarakshan et al., 2020). These generation models can generate unseen compounds in computers but do not answer how to synthesize them in practical. Retrosynthesis engines can help chemists and pharmacists fill this gap.

Practical retrosynthesis planning requires a strong model to learn inherent biases in the target dataset while keeping generalization performance to generate unseen (test) product compounds. The current trend is to focus on developing such a strong ML model architecture such as seq-to-seq models (Liu et al., 2017; Karpov et al., 2019) and graph-to-graphs models (Shi et al., 2020; Yan et al., 2020; Somnath et al., 2020), which achieve the State-of-the-Art (SotA) retrosynthesis accuracy.

However, model architecture is not the only issue to consider. In the current deep neural network (DNN) era, the quantity (many samples) and the quality (less noisy, corrupted samples) of the available dataset often governs the final performance of the ML model. The problem is that there are only a few large and high-quality supervised training datasets that are available publicly. Instead, only a small and/or a noisy dataset is usually available for the target application task. To cope with this problem, *data transfer* approaches are widely employed as ordinary research pipelines in computer vision (CV), natural language processing (NLP), and machine translation (MT) domains (Kornblith et al., 2019; Xie et al., 2020; He et al., 2020; Khan et al., 2019; Sennrich et al., 2016). A data transfer approach tries to transfer knowledge to help a difficult training on the small and/or noisy *target* dataset. That knowledge is imported from *augmented* dataset, which is usually *a large or*

$Y$: Reactant(s)

$p(X|Y)$: Synthesis

$p(X|Y)$: Retrosynthesis

$X$: Product(s)

Form. of $p(Y|X)$: Retrosynthesis model

Graph representations
(Somnath et al., 2020)
(Yan et al., 2020)
(Shi et al., 2020

SMILES representations
(Liu et al., 2017)
(Karpov et al., 2018)

CCOC(CN)=O.CC(CC1CCCC1=O)=O

CCOC(Cn1c(CCC2)c2cc1C)=O

Figure 1: Overview of the retrosynthesis problem.

*clean dataset in the same domain* but *does not share the same task or the same assumptions* with the target dataset. Such augmented datasets are beneficial if the quantity or the quality of the augmented dataset is superior compared to the target dataset. However, this data transfer approach is not still well investigated in the previous retrosynthesis studies as we explained above.

In this paper, we conduct a systematic investigation of the effect of data transfer for the improvement of the retrosynthesis models. We examine three standard methods of data transfer: joint training, self-training, and pre-training plus fine-tuning. The result shows that every data transfer method can improve the test prediction accuracy of an off-the-shelf Transformer retrosynthesis model. Especially, a Transformer with pre-training plus fine-tuning achieves comparable performance with, and in some cases better performance than, the SotA models. In addition, we conducted an intensive manual inspection for the erroneous prediction results. This inspection clarifies the limitations of our approaches. But at the same time, it reveals that the pre-training plus fine-tuning model can generate chemically appropriate or sensible proposals more than 99% cases of top-1 predictions.

## 2 RELATED WORKS

### 2.1 MACHINE LEARNING-BASED RETROSYNTHESIS

Computers are used to aid retrosynthesis design for decades, dating back to Corey & Wipke (1969). However, it takes many years to see the rise of generalizable and scalable ML-based approaches, instead of rule-based approaches (e.g. Chen & Baldi (2009)), which cannot extrapolate beyond the rules given, and first principle-based approaches (e.g. Wang et al. (2014), which suffer from prohibitively heavy computations.

The most recent ML retrosynthesis employs Deep Neural Network (DNN) for its main component (e.g. (Wei et al., 2016)). Among them, Liu et al. (2017) first introduced the LSTM seq-to-seq model. Their model handled the compounds with the SMILES (Weininger, 1988)-format string representation, and the retrosynthesis problem was solved as the MT problem. Later, Karpov et al. (2019) replaced the LSTM with the Transformer (Vaswani et al., 2017), which is the current baseline seq-to-seq DNN, and achieves a good performance.

Recently, Shi et al. (2020) introduced a graph-to-graphs approach. Graph-to-graphs approach can treat the compounds as molecular graphs with help of graph neural networks. This approach well matches human experts' intuition and the very recent model (Somnath et al., 2020) greatly improved the accuracy, outperforming the former State-of-the-Art (SotA) model based on the deep logic network (Dai et al., 2019).

As seen, most efforts of ML-based retrosynthesis researches are dedicated to stronger DNN architectures. In this paper, we propose another approach to improve the SotA of retrosynthesis, i.e., by transferring the knowledge of additional datasets that are not directly prepared for the users' target tasks.

### 2.2 DATA TRANSFER IN GENERAL ML DOMAINS

Generally, the performances of the statistical ML model are dependent on the design of the model (variable dependencies) and the datasets. In the dataset side, the both of the quality (fewer noises in features, less mislabels or conflicting samples) and the quantity (many sample sizes, not-few and

not-too-many numbers of class labels) of the training dataset are crucial. However, it is difficult, or practically impossible, to prepare a large dataset of high-quality samples for users' target tasks. To cope with this problem, *data transfer* techniques are widely employed as ordinary research pipelines in CV, NLP, and MT domains.

Pre-training is to train a model with some datasets *prior* to fine-tuning, which is further training with the target dataset. Sometimes the pre-trained model is deployed as it is as a general-purpose baseline. One of the first major successes of data transfer for DNN is the supervised pre-training in CV. In that circumstance, pre-trained binaries of successful networks (Krizhevsky et al., 2012; Simonyan & Zisserman, 2015; He et al., 2016) included in deep learning frameworks (e.g. Caffe) played crucial roles in advancing other CV studies. Today it is a common pipeline to employ the pre-trained model as a part of the larger and stronger networks (Girshick et al., 2014; Long et al., 2015; Xiong et al., 2018; Kornblith et al., 2019). One distinct characteristic of the pre-training in the NLP domain is the large sizes in the dataset to feed and the train models. Very large networks such as BERT (Devlin et al., 2018) and GPT-2 (Radford et al., 2019) are trained with massive amounts of the corpus (dataset), which were unrealistically huge for most engineers and researchers to deal with, were distributed in binaries or parameters. These pre-trained models were incorporated in each developer's system and contributed to push limits of several NLP studies and applications (e.g. (Lee et al., 2019; Lan et al., 2020; Yang et al., 2019)).

Self-training, another data transfer method studied in this paper, has a longer history than the pre-training, back to the seminal work by Scudder (1965). In the self-training approach, we first train a base model with the labeled target dataset, which is often small but high-quality. Then, a larger dataset that contains unlabeled or attached with low-quality noisy labels is *relabeled* by the base model. The relabeled dataset is used to augment the small target dataset to obtain a stronger model. The same technique is often referred to as *pseudo labeling*, one of the elemental techniques to boost performances in program competitions. Recently self-training draws attention in several domains, such as CV (Xie et al., 2020), NLP (He et al., 2020), and speech recognition (Khan et al., 2019). An advantage of self-training against the pre-training is that the self-training does not require the large labeled dataset thanks to the relabeling, even for the supervised task. This characteristic is employed in the MT application: it is difficult to collect qualified translation datasets (parallel corpus) for minor languages. Thus the self-training (referred to as *back translation*) is intensively used to augment the pseudo parallel corpus (Sennrich et al., 2016).

Zoph et al. (2020) examined the data transfer pipeline and showed that the self-training is superior to the widely adopted pre-training in object recognition (segmentation) tasks. They studied the data transfer at discriminative models. In this paper, we focus to examine the effectiveness of the data transfer methods in sequence-to-sequence generative models.

### 2.2.1 DATA TRANSFER IN BIO- AND CHEMO-INFORMATICS

A few researchers in bioinformatics and chemoinformatics show interest in the data transfer approach to learn a universal numerical representation (embedding) of properties of the compounds that is informative enough to predict multiple quantities. Honda et al. (2019) employed a Transformer encoder-decoder to pre-train the intermediate representation of compounds and applied the pre-trained encoder for downstream tasks combining with a separate predictor network. Li & Fourches (2020) followed the masked prediction procedure of BERT (Devlin et al., 2018) for pre-training. Wang et al. (2019) employed an LSTM but reports that the pre-trained model performs well in some small datasets. Compared to these previous studies, our study differs in the following two essential points. First, as we have explained earlier, we deal with data transfer for sequence-to-sequence generative models for retrosynthesis while these studies focus on discriminative models similar to (Zoph et al., 2020). Second, these previous studies did not perform systematic comparisons of *different transfer methods* for the target tasks. In contrast, we perform systematic and intensive comparisons among transfer methods to find the best data transfer methods for retrosynthesis.

There are two exceptions that study the transfer learning in generative models in chemoinformatics. Pesciullesi et al. (2020) improved accuracy of synthesis prediction by applying the Molecular Transformer model to carbohydrate reactions using transfer learning. Though apparently similar, the synthesis and the retrosynthesis belong to different classes of problems: a set of reactants yields a specific product compound while a product can be decomposed into multiple reactant sets.

Chen et al. (2019) proposed to pre-train a retrosynthesis model in a slightly different way. They *synthesized* a pre-training dataset by manipulating the *target dataset* randomly or using pre-defined rules and did not rely on the other large datasets. They reported such pre-training was still able to improve retrosynthesis accuracy, though the gains were not large. Their problem setting was different from ours. We focused on the case where an additional huge dataset is available other than the target dataset to mitigate the quantity-and-quality problem. We will show that our approach can achieve much greater improvements in retrosynthesis accuracy.

## 3 DATA TRANSFER METHODS FOR RETROSYNTHESIS

In this section, we first give a general formulation of ML-based retrosynthesis problem. Then, we explain several data transfer methods using that formulation.

### 3.1 RETROSYNTHESIS PROBLEM FORMULATION

Retrosynthesis is a problem to map a product compound item $x$ to an appropriate set $y$ of reactant compounds (Figure1). A sample for a retrosynthesis model is a pair $(x, y)$ where $x$ denotes a product, $y$ denotes a set of reactants. A retrosynthesis model $\mathcal{M}$ is a (stochastic) mapping between $x$ and $y$: conventionally formulated as a likelihood distribution $p_{\mathcal{M}}(y|x; \theta)$ with the parameters $\theta_{\mathcal{M}}$. Formulation of the likelihood determines the model characteristics. Most previous studies focused on developing new sophisticated formulations e.g. seq-to-seq models (Liu et al., 2017; Karpov et al., 2019) and graph-to-graphs models (Shi et al., 2020; Somnath et al., 2020; Yan et al., 2020).

In this paper, we simply set $\mathcal{M}$ a naive seq-to-seq Transformer (Vaswani et al., 2017) since we focus on the data transfer rather than the model architecture. Now the model is fixed, so we omit $\mathcal{M}$ hereafter. Under a seq-to-seq formulation of retrosynthesis, both of the product $x$ and the set of reactants $y$ are represented as strings (a sequence of characters). We adopt the standard string representation of chemical compounds, referred to as SMILES (Weininger, 1988). Basically any compound can be uniquely represented as a SMILES format string. $y$ consists of possibly multiple compounds. In such a case, we simply concatenate the multiple SMILES strings into a longer string with delimiters.

Assume a dataset $\mathcal{D}$ is a set of reaction samples where the product and the reactants are represented as SMILES strings. The $i$th sample of the dataset is indexed by $i$: a pair $(x_i, y_i)$ thus $\mathcal{D} = \{(x_i, y_i)\}$. A training dataset $\mathcal{D}_{\text{train}}$ is used to find good parameters $\theta_{\mathcal{M}*}$ that optimize an objective function, which is typically the maximum (log-)likelihood criterion:

$$\theta^* = \arg_\theta \max \sum_{(x_i, y_i) \in \mathcal{D}_{\text{Train}}} \log p(y_i|x_i) . \tag{1}$$

We usually implement the above maximum likelihood criterion as the minimization of the cross-entropy loss $\mathcal{L}(\mathcal{D}_{\text{Train}})$. We minimize the cross-entropy loss by an iterative parameter update algorithm such as stochastic gradient descent:

$$\theta^0 \leftarrow \text{Init}, \quad \theta^{\ell+1} \leftarrow \theta^\ell - \gamma^\ell \nabla \mathcal{L}(\mathcal{D}_{\text{Train}})|_{\theta^\ell} . \tag{2}$$

The iterations are scheduled by validation scores computed on a validation dataset $\mathcal{D}_{\text{val}}$, and the learned model is evaluated by test scores computed on a test dataset $\mathcal{D}_{\text{test}}$. These three datasets should have no intersection samples.

### 3.1.1 BASELINE: SINGLE MODEL TRAINING (WITHOUT DATA TRANSFER)

The standard setting of most of the previous retrosynthesis models followed a simple scenario. First, we prepare a dataset to which we want to fit the model. We refer to this dataset as a **target dataset** $\mathcal{D}^T$. The whole $\mathcal{D}^{\mathcal{T}}$ is split into three sub-datasets: a training set $\mathcal{D}_{\text{Train}}^T$, a validation set $\mathcal{D}_{\text{Val}}^T$, and a test set $\mathcal{D}_{\text{Test}}^T$. The single model training is:

$$\theta_{\text{single}}^* = \arg_\theta \max \sum_{(x_i, y_i) \in \mathcal{D}_{\text{Train}}^T} \log p(y_i|x_i) , \tag{3}$$

where the training procedure is scheduled by $\mathcal{D}_{\text{Val}}^T$ and evaluated on $\mathcal{D}_{\text{Test}}^T$. In this setting, we have no dataset other than the target dataset, thus we do not (can not) conduct any data transfer.

### 3.2 Data Transfer for Retrosynthesis

Next, we consider another scenario where we are given an additional dataset other than the target dataset. We call this additional dataset as **augment dataset** $\mathcal{D}^A$. Typically augment datasets have larger sample sizes than those of the target datasets. It is also usual that empirical distributions of $p(y|x)$ can be different between $\mathcal{D}^T$ and $\mathcal{D}^A$. Still, we want to *transfer* some knowledge from $\mathcal{D}^A$ to the model parameter to improve **the test scores computed on the target test set** $\mathcal{D}_{\text{test}}^T$.

#### 3.2.1 Joint Training

One of the most simple ways of data transfer is joint training. We split $\mathcal{D}^A$ into a training set $\mathcal{D}_{\text{Tran}}^A$, a validation set $\mathcal{D}_{\text{VAl}}^A$, and a test set $\mathcal{D}_{\text{Test}}^A$.

In the joint training, we optimize the parameters that satisfy the training objective on both $\mathcal{D}_{\text{Train}}^T$ and $\mathcal{D}_{\text{Train}}^A$. In our setting, the model and the objective are shared among datasets, so we simply train on the concatenated training sets:

$$\theta_{\text{joint}}^* = \arg_\theta \max \sum_{(x_i, y_i) \in \mathcal{D}_{\text{joint}}} \log p\left(y_i | x_i\right) , \text{ where } \mathcal{D}_{\text{joint}} = \mathcal{D}_{\text{Train}}^T \cup \mathcal{D}_{\text{Train}}^A . \tag{4}$$

The training process is scheduled by $\mathcal{D}_{\text{Val}}^T$ and evaluated on $\mathcal{D}_{\text{Test}}^T$.

The above formulation (Eq.4) of the joint training is restrictive: only available when the domain of the data pair $(x, y)$ are shared among the target dataset and the augment dataset. For example, in our cases, the canonicalization rule of the SMILES compound string representation must be the same in $x$ $(y)$ of $\mathcal{D}_{\text{Train}}^T$ and $\mathcal{D}_{\text{Train}}^A$.

### 3.3 Self-Training

Self-training is a well-known technique for data transfer, and it is also known as *pseudo labeling*. Unlike the joint training, we can conduct the self-training even if the domain of $y$ differs between $\mathcal{D}_{\text{Train}}^T$ and $\mathcal{D}_{\text{Train}}^A$.

We first conduct the single model training (using $\mathcal{D}^T$ solely) and obtain $\theta_{\text{single}}^*$. Then, we *decode all product of the augment dataset* with the learned parameter $\theta_{\text{single}}^*$. The generated *pseudo labels* $\hat{y}$ is used to replace the reactants of the new augment training set. Then we perform the joint training by concatenating $\mathcal{D}_{\text{Train}}^T$ and the pseudo-labeled $\mathcal{D}_{\text{Train}}^A$. Decoding with the single model trained by the target dataset will ease the difficulty of transferring knowledge due to the inconsistent $p(y|x)$ between $\mathcal{D}^T$ and $\mathcal{D}^A$. We can formulate the self-training as follows:

$$\hat{y}_i = \arg_y \max \log p\left(y | x_i; \theta_{\text{single}}^*\right) \text{ for } x_i \in \mathcal{D}_{\text{Train}}^A , \tag{5}$$

$$\mathcal{D}_{\text{PseudoTrain}}^A = \{(x_i, \hat{y}_i)\} \text{ for all } x_i \in \mathcal{D}_{\text{Train}}^A . \tag{6}$$

$$\theta_{\text{self}}^* = \arg_\theta \max \sum_{(x_i, y_i) \in \mathcal{D}_{\text{self}}} \log p\left(y_i | x_i\right) , \text{ where } \mathcal{D}_{\text{self}} = \mathcal{D}_{\text{Train}}^T \cup \mathcal{D}_{\text{PseudoTrain}}^A . \tag{7}$$

The training process of Eq.7 is scheduled by $\mathcal{D}_{\text{Val}}^T$ and evaluated on $\mathcal{D}_{\text{Test}}^T$.

### 3.4 Pre-training plus Fine-tuning

In the joint training and the self-training, the final best parameter is computed via the training on the concatenated training sets $\mathcal{D}_{\text{joint}}$ and $\mathcal{D}_{\text{self}}$. The sample sizes of these concatenated training sets can be large, resulting in slower training convergence. Pre-training plus fine-tuning method handles this problem by training a model in advance with the augment dataset $\mathcal{D}^A$. The *pre-trained model* is simply loaded as the initial model, and is *tuned* to the smaller target dataset. The later *fine-tuning* finishes faster in general, beneficial for the cases when we have potentially multiple target datasets for deployments.

The pre-training plus fine-tuning approach thus naturally requires two training phases. In the pre-training phase (the first phase), we preform a single model training **solely on the augment dataset:**

$$\theta^*_{\text{pretrain}} = \arg_\theta \max \sum_{(x_i,y_i)\in \mathcal{D}^A_{\text{Train}}} \log p\left(y_i|x_i\right) , \tag{8}$$

where the training procedure is scheduled by $\mathcal{D}^A_{\text{Val}}$ and evaluated on $\mathcal{D}^A_{\text{Test}}$.

In the fine-tuning phase, we train the model with $\mathcal{D}^T$ not from the scratch but from $\mathcal{M}_{\theta^*_{\text{pretrain}}}$.

$$\theta^0_{\text{finetune}} \leftarrow \theta^*_{\text{pretrain}} , \quad \theta^{\ell+1}_{\text{finetune}} \leftarrow \theta^\ell_{\text{finetune}} - \gamma^\ell \nabla \mathcal{L}(\mathcal{D}^T_{\text{Train}})|_{\theta^\ell_{\text{finetune}}} , \tag{9}$$

$$\theta^*_{\text{finetune}} \leftarrow \theta^{\ell\to\infty}_{\text{finetune}} . \tag{10}$$

where the training procedure is scheduled by $\mathcal{D}^T_{\text{Val}}$ and evaluated on $\mathcal{D}^T_{\text{Test}}$.

## 4 EXPERIMENT

In this section, we first assess the effects of the several data transfer methods, compared with the Transformer that is trained naively. After that, we show the comparisons with the most recent SotA retrosynthesis models.

### 4.1 PROCEDURE

#### 4.1.1 DATASET

Most ML-based retrosynthesis studies adopt the USPTO database, which is a collection of chemical reaction formulas automatically OCR-scanned from the US Patent documents.

The target dataset is the USPTO-50K dataset, which contains curated 50K samples provided by Lowe (2012). This dataset serves as the standard benchmark in the retrosynthesis studies. The characteristic of this dataset is that all samples are classified in one of ten major reaction classes in advance (Schneider et al., 2016). This means the distribution of products and reactants are skewed. In this study, the reaction class information is only used to summarize the prediction results and is not used for filtering the dataset or for prediction.

For data transfer, we prepare two augment datasets. Since these two datasets are not filtered based on the reaction classes, their distributions of compounds are clearly different from that of the USPTO-50K dataset. The first and the main augment dataset is referred to as USPTO-Full dataset, which contains 877K samples curated by Lowe (2017). The second augment dataset is referred to as USPTO-MIT dataset, which contains 479K samples curated by Jin et al. (2017). Experimental results on USPTO-MIT is included in the appendix for sample sizes assessment.

The three datasets have a number of shared reaction samples or noisy instances. For fairer evaluations, we need data cleansing beforehand. The details of the data cleansing are presented in the appendix.

#### 4.1.2 MODEL ARCHITECTURE, TRAINING, AND EVALUATIONS

In order to give fair assessments of the effects of data transfer methods, we fix the architecture of the seq-to-seq model, namely the Transformer Vaswani et al. (2017) regardless of the transfer methods. We may obtain better results if we optimize the architecture for each transfer method.

All models are optimized via Adam (Kingma & Ba, 2015). Following the seminal paper of Karpov et al. (2019), learning rates are scheduled in a cyclic manner with the warm-up iterations excepting the fine-tune training. In the fine-tuning, we find that a standard non-cyclic scheduler (Klein et al., 2017) perform good, thus we adopt it in our experiments.

We adopt the $n$-best accuracy score of the predicted results on the test set as the evaluation metric. We use $k = 50$ beam search to list and sort the predictions, and compute $n = 1, 3, 5, 10, 20, 50$-best accuracy scores, following the normal procedure in the literature.

Previous works (Dai et al., 2019; Somnath et al., 2020; Yan et al., 2020) use the validation sets to exponentially decay the learning rate. Instead, we use the validation set to choose and output

Table 1: $n$-best accuracy of retrosynthesis tasks on USPTO-50K, with different data-transfer training methods. Augment dataset is the cleansed USPTO-Full. Larger values are better. Averages and standard deviations of 5 runs are presented. Bold faces indicate the best scores at the specific $n$-accuracy.

| Training Method | $n$-best accuracy (%) | | | | | |
|---|---|---|---|---|---|---|
| | $n=1$ | $n=3$ | $n=5$ | $n=10$ | $n=20$ | $n=50$ |
| Single model (No Transfer) | $35.3 \pm 1.4$ | $52.8 \pm 1.4$ | $58.9 \pm 1.3$ | $64.5 \pm 1.2$ | $68.8 \pm 1.2$ | $72.1 \pm 1.3$ |
| Joint Training | $39.1 \pm 1.3$ | $63.4 \pm 0.9$ | $71.9 \pm 0.5$ | $80.1 \pm 0.2$ | $85.4 \pm 0.3$ | $89.4 \pm 0.2$ |
| Self-Training | $41.5 \pm 1.0$ | $60.4 \pm 0.7$ | $66.1 \pm 0.7$ | $71.8 \pm 0.6$ | $75.3 \pm 0.5$ | $78.0 \pm 0.3$ |
| Pre-training + Fine-Tune | $\mathbf{57.4} \pm 0.4$ | $\mathbf{77.6} \pm 0.4$ | $\mathbf{83.1} \pm 0.2$ | $\mathbf{87.4} \pm 0.4$ | $\mathbf{89.6} \pm 0.3$ | $\mathbf{90.9} \pm 0.2$ |

a snapshot that records *the best validation perplexity*. All accuracy scores of our experiments are computed on these best val-score snapshots.

Full descriptions about the experimental procedure are presented in the appendix.

## 4.2 RESULTS

### 4.2.1 COMPARING DATA TRANSFER METHODS

Table 1 shows the $n$-best accuracy scores over several data transfer methods. We use the USPTO-50K dataset as the target dataset, and the cleansed USPTO-Full dataset as the augment dataset in this table.

Generally, all data transfer methods are successful in improving the $n$-best accuracy for all choices of $n$. Among them, the pre-training plus fine-tuning method achieves the remarkable gains (more than 20 points for most $n$). This result is indeed beneficial for the researchers who conduct not computational experiments but actual synthesis experiments because it reduces the number of required experiment trials. In this study, we did not perform optimization of the model itself because our main goal was to confirm the effect of data transfer, but if necessary, we can perform optimization with the existing model immediately.

It is interesting to compare this result with the previous study by Zoph et al. (2020), which examines the self-training and the pre-training plus fine-tuning in the image recognition tasks. Their conclusion is that pre-training is less effective than the self-training to improve generalization performance. The authors explain this is because there is no *universal representation (embedding)* of generic images. In this study, which is not for image recognition but for retrosynthesis, we observe that the pre-training plus fine-tuning is evidently better than the self-training. Perhaps the chemical compound strings may have a universal representation because of restrictions and patterns naturally imposed on the chemical compounds in nature.

### 4.2.2 COMPARISONS WITH THE STATE-OF-THE-ARTS

Next, we compare our best data transfer model with the SotA model of the retrosynthesis. As the figure shows, our best pre-training plus fine-tuning model performs as good as the latest SotA model in the literature, which is built on a off-the-shelf simple Transformer. Moreover, our model exceeds the known best accuracy at $n = 10$ and $n = 20$.

None of the compared studies provided the standard deviations of the obtained accuracy scores. Thus it is difficult to conclude definitively that one model is better than other models from this result. However, our model performed the best in $n = 10, 20$-best accuracy, which is preferable for those who conduct synthesis experiments, for many reasons. First, there is usually more than one correct answer for retrosynthesis analysis, even if it consists of only one-step reaction. Second, the reaction with the highest score is not always the best in practice. Thus, it is expected to output various solutions within a limited number (for example, 10) of answers rather than producing only one answer. In other words, a predictor that makes 10 good suggestions is pragmatically more highly valued than a predictor that makes only one suggestion that is considered the best.

Table 2: $n$-best accuracy of retrosynthesis tasks on USPTO-50K, comparing with the most recent SotA models. Bold faces indicate the best scores at the specific $n$-accuracy. All numbers of the compared models are borrowed from their original papers. N/A indicates the accuracy scores are not reported in the published papers.

| Name | Model Arch. | $n$-best accuracy (%) | | | | | |
|---|---|---|---|---|---|---|---|
| | | $n=1$ | $n=3$ | $n=5$ | $n=10$ | $n=20$ | $n=50$ |
| GLN (Dai et al., 2019) | Logic Network | 52.5 | 69.0 | 75.6 | 83.7 | 88.5 | **92.4** |
| G2Gs (Shi et al., 2020) | Graph-to-Graph | 48.9 | 67.6 | 72.5 | 75.5 | N/A | N/A |
| RetroXpert (Yan et al., 2020) | Graph-to-Graph | **65.6** | 78.7 | 80.8 | 83.3 | 84.6 | 86.0 |
| GraphRetro (Somnath et al., 2020) | Graph-to-Graph | 63.8 | **80.5** | **84.1** | 85.9 | N/A | 87.2 |
| Our Pre-training + Fine-Tune | Seq-to-Seq | 57.4 | 77.6 | 83.1 | **87.4** | **89.6** | 90.9 |

### 4.3 MORE ANALYSES

Due to page limitation, we describe other additional analyses in appendixes. Here we only summarize the main messages of these analyses. Please find the appendix for the details.

1. Additional experiments with smaller USPTO-MIT dataset shows that the sample size of the augment dataset clearly affects the generalization performance of the transferred models.

2. We find that the 1-best accuracy and $n$-best accuracy show quite different evolution curves against training iterations, especially for the single training model. Such behaviors are not mentioned in the literature so far, possibly opening a new question for model training. This behavior is not observed in the pre-training plus fine-tuning, which is another advantage over other transfer methods.

3. We confirm that the pre-training plus fine-tuning improves the class-wise prediction accuracy for all 10 major reaction classes of USPTO-50K. The augment dataset especially help prediction of difficult bond/ring formation reactions.

4. The manual inspection for the erroneous prediction results reveals that there are a few reactions that are still difficult to perform reasonable retrosynthesis prediction.

5. At the same time, the manual inspection results show that over 99% of top-1 predictions are chemically reasonable and appropriate hypotheses, even if the hypotheses do not include the exact "gold" reactant hypothesis.

## 5 CONCLUSION

In this paper, we conducted a systematic investigation on the effect of data transfer for the improvement of the computational retrosynthesis system. The result proved that the most typical data transfer methods can improve the test prediction accuracy of a retrosynthesis model. Especially, a Transformer with pre-training plus fine-tuning updated the SotA of the 10- and 20-best retrosynthesis accuracy. We also confirmed that the pre-training plus fine-tuning model can generate chemically appropriate or sensible proposals more than 99% cases of top-1 predictions through our manual inspections of the predictions.

We only validated the simplest transfer techniques in this work. As future work, we are interested in applying more sophisticated transfer learning methods. For example, freezing a part of the NN during fine-tuning may be effective according to the results in CV domain. It is also interesting for the retrosynthesis community to apply the data transfer methods for graph-to-graphs models such as (Somnath et al., 2020). We observed that the USPTO-Full dataset is transferable to the USPTO-50K dataset, which has a biased distribution of reaction types. To examine the transferability of the USPTO-Full dataset, we need to test the transfers to other retrosynthesis datasets, which are small, biased and collected for specific purposes.

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

## APPENDIX A   EXPERIMENT PROCEDURE: DETAILS

### A.1   DATASET

Most ML-based retrosynthesis studies adopt the USPTO database, which is a collection of chemical reaction formulas automatically OCR-scanned from the US Patent documents. We use the filtered subsets of the USPTO database in this study.

The target dataset is the USPTO-50K dataset, which contains curated 50K samples provided by Lowe (2012). This dataset serves as the standard benchmark in the retrosynthesis studies. The dataset is split into 40K train, 5K validation, and 5K test sets following (Liu et al., 2017). The distinct characteristic of this dataset is that all 50K samples are classified in one of ten major reaction classes in advance (Schneider et al., 2016). This means the distribution of products and reactants are highly skewed. In this study, the reaction class information is only used to summarize the prediction results and is not used for filtering the dataset or for prediction.

To perform data transfer, we prepare two augment datasets. Since these two datasets are not filtered based on the reaction classes, their distributions of compounds are clearly different from that of the USPTO-50K dataset. The first and the main augment dataset is referred to as USPTO-Full dataset, which is curated by Lowe (2017). The USPTO-Full dataset is split into 879K, 49K, and 49K samples for training, validation, and test sets, respectively. The second augment dataset is referred to as USPTO-MIT dataset, which is curated by Jin et al. (2017). The USPTO-MIT dataset is split into 409K, 30K, and 40K samples for training, validation, and test sets, respectively[1]. Experimental results on this dataset are presented in the appendix for assessment of the sample sizes of augment datasets.

### A.2   DATA CLEANSING

The training sets of the augment datasets (USPTO-Full and USPTO-MIT) contain many reaction SMILES samples that are incorporated in one of the subsets of the target dataset USPTO-50K. Such contaminated samples inhibit fair predictions, because of the duplicated training samples (duplicated samples given more weight than other samples) or the data leaks (if a test sample is included in the training set, the model can answer correctly by simply memorizing the sample without learning the features). Thus, we remove all reaction samples from the training sets of the two augment datasets whose *product* SMILES are included in any subsets of the target dataset. As a result, the USPTO-MIT training set is reduced to 384K samples. Similarly, the USPTO-Full training set shrinks to 844K samples.

We adopt a canonical SMILES format throughout the experiments. However, there are multiple canonicalization rules in this field. In fact, we confirmed the canonicalization rules are not unified among the three datasets. Non-unified canonicalizations may hinder effective training, and may also induce unexpected data leaks because the same compound with different SMILES string cannot be detected. Therefore we conducted unified canonicalization for all datasets using the RDKit tool (Landrum & others, 2006) with a specific version.

### A.3   MODEL ARCHITECTURE, TRAINING, AND EVALUATIONS

In order to give fair assessments of the effects of data transfer methods, we fix the architecture of the seq-to-seq model, namely the Transformer Vaswani et al. (2017) regardless of the transfer methods. This means the reported results are based on the sub-optimal Transformer architecture; we may obtain better results if we optimize the architecture for each transfer method. The number of self-attention layers is set to 3, and the dimensions of the latent vectors are set to 500 in all layers. We adopt the positional encoding, which we find it essential to achieve good scores through preliminary experiments. We limit the maximum number of tokens (length of a sequence) up to 200 to avoid memory shortage of GPUs. We use the off-the-shelf implementation of the Transformer by OpenNMT-py[2].

---

[1]Downloaded from https://github.com/wengong-jin/nips17-rexgen/blob/master/USPTO/data.zip
[2]opennemt.org

Table 3: $n$-best accuracy of retrosynthesis tasks on USPTO-50K, with different data-transfer training methods. Augment dataset is the smaller USPTO-MIT. Other details are the same with the Table 1.

| Training Method | $n$-best accuracy (%) | | | | | |
|---|---|---|---|---|---|---|
| | $n=1$ | $n=3$ | $n=5$ | $n=10$ | $n=20$ | $n=50$ |
| Single model (No Transfer) | $35.3 \pm 1.4$ | $52.8 \pm 1.4$ | $58.9 \pm 1.3$ | $64.5 \pm 1.2$ | $68.8 \pm 1.2$ | $72.1 \pm 1.3$ |
| Joint Training | $38.4 \pm 0.9$ | $60.7 \pm 0.5$ | $67.8 \pm 0.4$ | $75.2 \pm 0.3$ | $80.4 \pm 0.4$ | $84.9 \pm 0.3$ |
| Self-Training | $41.2 \pm 0.3$ | $60.2 \pm 0.4$ | $66.2 \pm 0.2$ | $71.9 \pm 0.3$ | $75.5 \pm 0.5$ | $78.2 \pm 0.5$ |
| Pre-training + Fine-Tune | $\mathbf{52.2} \pm 0.4$ | $\mathbf{73.1} \pm 0.4$ | $\mathbf{78.8} \pm 0.4$ | $\mathbf{83.7} \pm 0.3$ | $\mathbf{86.3} \pm 0.3$ | $\mathbf{88.2} \pm 0.3$ |

All models are optimized via Adam (Kingma & Ba, 2015). Following the seminal paper of Karpov et al. (2019), learning rates are scheduled in a cyclic manner with the warm-up iterations excepting the fine-tune training. In the fine-tuning, we find that a standard non-cyclic scheduler (Klein et al., 2017) performs good, thus we adopt it in our experiments.

We adopt the $n$-best accuracy score of the predicted results on the test set as the evaluation metric. Remember that a sample is a pair of the input product SMILES string and the expected reactant SMILES string. However, the provided reactant SMILES is not necessarily the only correct answer. Thus we predict the $k$-best possible predictions for each test input. If the expected reactant SMILES is included in the best-$n$ prediction, we assume the retrosynthesis model is successful in predicting the answer. We use $k = 50$ beam search to list and sort the predictions, and compute $n = 1, 3, 5, 10, 20, 50$-best accuracy scores, following the normal procedure in the literature.

We note that there are possible choices of how to employ validation split samples. Previous works (Dai et al., 2019; Somnath et al., 2020; Yan et al., 2020) use the validation sets to exponentially decay the learning rate. However, we adopt the cyclic learning rate scheduler following the seminal Transformer-based model (Karpov et al., 2019) and we found it indeed effective in our experiments. Therefore, we use the validation set to choose and output a snapshot that records *the best validation perplexity*. All accuracy scores of our experiments are computed on these best validation score snapshots. We do not employ any ensemble model for fair comparisons with the previous studies.

## APPENDIX B    SMALLER AUGMENT DATASET RESULTS IN INFERIOR PERFORMANCE

Table 3 shows the $n$-best accuracy scores over several data transfer methods, but this time we use the smaller USPTO-MIT dataset as the augment dataset in this table. Our expectation is the gains of data transfer methods decreases with the smaller augment dataset.

The joint training and the pre-training plus fine-tuning record worse scores compared to the USPTO-Full cases (Table 1. Surprisingly, the gain of the self-training is not affected by the sample size of the augment dataset.

## APPENDIX C    TRAINING EVOLUTION AND TEST GENERALIZATION PERFORMANCE IN DIFFERENT TRANSFER METHODS

We thoroughly trained several models with larger numbers of mini-batch iterations (or epochs), which has not been conducted in any previous studies.

Figure 2 presents how the three metrics (the average values of five runs) on the target dataset (train and test sets) change as the training iteration increases. Spikes indicate the periods of the cyclic learning rate schedulers. Figure 2 consists of 4x3 panels, where the four rows correspond to four different transfer training methods: single model (no transfer), joint training, self-training, and pre-training plus fine-tuning[3], and the three columns correspond to the three different metrics. The left column shows

---

[3]We only consider the fine-tuning because we have no access to the target dataset during pre-training in usual cases.

the perplexity on the training set. We observe all transfer training methods successfully decrease the training perplexity, namely, better fit for the training set.

The middle and the right columns show the 1-best and the 20-best accuracy scores on the test set, respectively. The remarkable difference is observed among different models, especially between the single (1st row) and the self-training (3rd row) models. More importantly, this analysis clearly showed different behavior between 1-best and the $n(> 1)$-best accuracy curves in the single model.

One may naturally expect and hypothesize that 1-best and the $n(> 1)$-best accuracy curves behave similarly: but no previous retrosynthesis studies have examined, or even discussed, this hypothesis. We showed in this study that the hypothesis does not hold for single model training on USPTO-50K, which is the default training strategy in previous studies.

This observation, that the 1-best and the n-best accuracy curves may be totally different, is important especially when trying to verify the top-n result experimentally (not in a computational experiment, but in an actual experiment that is both time-consuming and expensive). One possible reason for this difference between the 1-best and the $n$-best accuracy is the form of the objective function. The maximum likelihood of objective $\mathcal{L}$ updates the parameter $\theta$ so as to maximize the 1-best accuracy on the training set, not to maximize the $n$-best accuracy.

It is also remarkable that the two curves of the pre-training plus fine-tuning (4th row) are not significantly different, but rather very similar. The trained model quickly hits the peaks of the 1-best and the 20-best accuracy around 10K iterations. After hitting the peaks, two curves decrease rapidly. The model gradually *forgets* the beneficial knowledge transferred from the augment dataset by keeping the training iterations, and the model will converge to the single train model after many iterations. Thus it is important whether we can detect the peaks to early-stop the fine-tuning. In our experiments, the validation score monitoring are always successful in identifying this peak, yielding the good scores for all top-$n$ accuracy in Tables. 1,3. The fact that the two curves of the pre-training plus fine-tuning approach are very similar is another advantage for the data transfer in retrosynthesis. The joint training (2nd row) also shows the similar curves in test scores, but the pre-training plus fine-tuning approach is better in two aspects: final accuracy scores (Table.1) and necessary numbers of iterations (roughly between 70K and 250K iterations) required to achieve the best-val-score snapshot.

### C.1 CONCERNING THE SINGLE MODEL SCORE

It may seem strange that the 1,3,5-best accuracy scores of the single model are lower than those of the known results in (Karpov et al., 2019)[4].

In our experiment, the best-val-score snapshots of the single model are always chosen from the earlier iterations (less than 10,000 iterations), resulting in low 1-best scores of the single model compared to the known results (Karpov et al., 2019). According to Figure 2. the 1-best accuracy exceeds the known score if we choose the snapshot of e.g. 140,000 iterations. However, the snapshot achieves worse 20-best accuracy compared to the best-val-score snapshot.

## APPENDIX D    MORE IN-DEPTH EXAMINATION OF PREDICTED REACTANTS

Next, we examine the predicted reactant SMILES to see how the data transfer contributes to the improvement of the computational retrosynthesis prediction from the perspective of actual chemical experiment. We evaluate 428 test samples $(x_j, y_j) \in \mathcal{D}_{\text{Test}}^T$ whose 50-best predictions $\hat{y}_j$s do not include the "gold" reactant SMILES string $y_j$. We note again that the gold reactant set $y_j$ is not the sole correct retrosynthesis prediction, but the current studies adopt this "hard" criterion. During the inspection, we realize that USPTO-50K test dataset still contains a number of errors (mislabels) within the test set despite the curating efforts of the original authors (Lowe, 2012). Therefore we exclude these mislabeled samples from this in-depth analysis.

### D.1    SUCCESSFUL TRANSFER CASES

Table 4 shows the accuracy of the single training model and our best-transferred model (pre-training plus fine-tuning) for each reaction class. From the table, we observe the heterocycle formation is

---

[4]The most similar model $T1_1$ achieved 39.8, 59.1, and 63.9, respectively

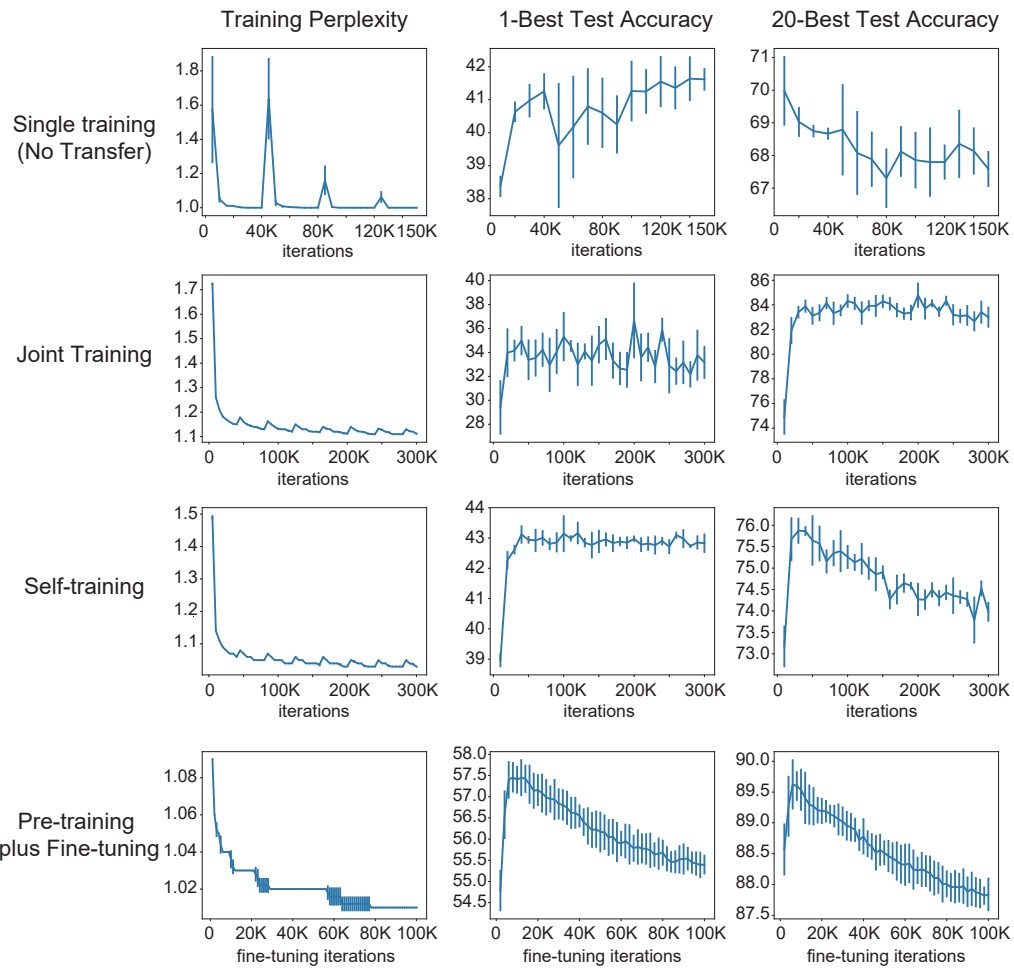

Figure 2: Relationships between Training iterations and Train/Test scores. For all panels, the horizontal axis denotes the training iterations. First row: Single model, Second row: joint training, Third row: Self-training, Fourth row: Pre-training plus Fine-tuning. The left column shows the training perplexities. The middle column shows the 1-best test accuracy scores. The right column shows the 20-best test accuracy scores.

difficult to predict with a single training of a seq-to-seq model (Liu et al., 2017). The heterocycle formation reaction not only forms some new bonds but also aromatizes some atoms through a reaction, which changes many capital letters in SMILES string to lowercase. Therefore, it is expected that it will be difficult to learn atom-to-atom mapping with small training samples. Figure 3 shows examples that fail in the single model but succeed in the transfer model. Significant improvements are observed in the prediction of complicated reactions such as heterocyclic reactions (1st row), suggesting that pre-training plus fine-tuning with a large augment dataset enables to learn such large changes. Deals-Alder reaction (2nd row) is another example of complicated reactions, where two C-C bonds are formed as well as some atom and bond types change. In general, the single model is not good at generating the ring formation reactions and only returns odd answers, while the transferred model is able to return correct answers to such reactions.

Table 4: Difference in 50-best accuracy for each reaction class.

| Reaction class | | Number of samples | 50-best accuracy | |
|---|---|---|---|---|
| | | | Single model | Transferred model |
| Heteroatom alkylation/arylation | RX_1 | 1497 | 0.78 | 0.96 |
| Acylation and related processes | RX_2 | 1176 | 0.85 | 0.98 |
| C-C bond formation | RX_3 | 553 | 0.54 | 0.79 |
| Heterocycle formation | RX_4 | 90 | 0.40 | 0.86 |
| Protection | RX_5 | 64 | 0.64 | 0.83 |
| Deprotection | RX_6 | 812 | 0.69 | 0.88 |
| Reduction | RX_7 | 457 | 0.77 | 0.96 |
| Oxidation | RX_8 | 81 | 0.77 | 0.93 |
| Functional group interconversion | RX_9 | 176 | 0.57 | 0.90 |
| Functional group addition | RX_10 | 22 | 0.55 | 0.86 |

Figure 3: Examples of being able to make predictions with pre-training and fine-tuning. The top-1 predictions for the single model are reasonable but the model does not predict reactants for constructing a ring. The 1st row (**a**) An example of heterocycle formation (RX_4). Synthesis *N*-substituted pyrrole from 1,4-diketone and alkylamine. Single model does not correctly propose reactants built a pyrrole ring. The 2nd row (**b**) An example of C-C bond formation (RX_3). Synthetic organic chemists easily come up with the Diels-Alder reaction when they see bicyclo[2.2.1]hept-2-ene ring system but single model does not predict any Diels-Alder reaction within the top-50.

## D.2  DIFFICULT CASES

Figure 4 shows difficult examples that could not be predicted correctly even by our best model. If there are multiple similar substituents in a compound, the transferred model sometimes chooses them wrong. In another case, our model fails to generate valid SMILES string. This indicates that the augment dataset still does not contain a sufficient amount of reactions for polycyclic aromatic hydrocarbons. We need more data augmentation to prevent that from happening, but augmentation may be difficult to do as such chemical groups are rare.

## D.3  MOST PREDICTIONS ARE CHEMICALLY APPROPRIATE, THOUGH NOT EXACTLY MATCH THE GOLD ANSWER

Having confirmed the predicted results based on our expertise in synthetic organic chemistry, less than 0.5% of the top 1 reactions were found to be wrong, and less than 0.2% of the cases did not output any organically correct reactions at all. This is a difficulty of the evaluation of retrosynthesis: there are multiple reasonable (appropriate) hypotheses for reactant predictions, and the n-best accuracy does not perfectly match the problem. At the same time, it is surprising that our model achieves such high accuracy without using domain knowledge or graph representation of compounds.

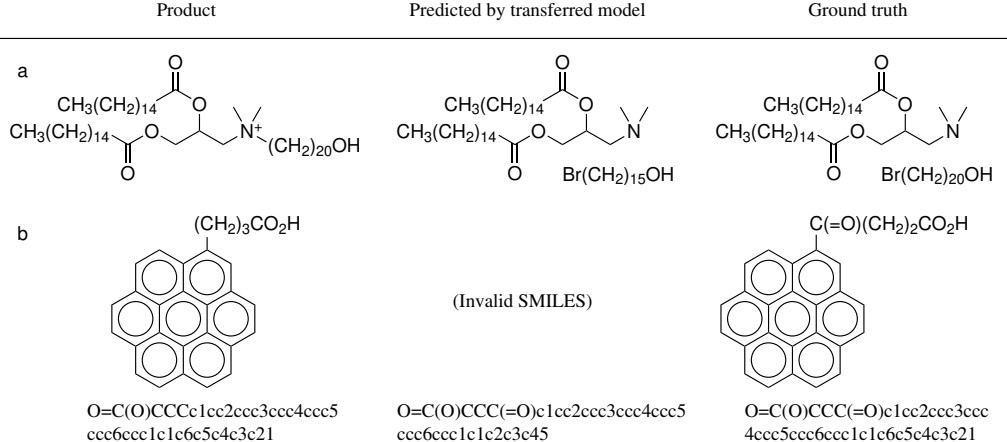

Figure 4: Examples of failed predictions. The 1st row (**a**). The model fails when there are multiple similar substituents like long chain hydrocarbons. The 2nd row(**b**) Our model can predict Clemmensen reduction reactions, but outputs incorrect SMILES substrings corresponding to the coronene substructure.

