# OpenReview forum: "Data Transfer Approaches to Improve Seq-to-Seq Retrosynthesis"
_ICLR.cc/2021/Conference — Reject_

### Official Review · AnonReviewer4 · 2020-10-20

**Rating:** 4
**Confidence:** 4

**Review:**

Summary & Strengths
- Recent works for retrosynthesis focus on developing neural architectures in the USPTO-50 benchmark. In contrast, this paper aims at improving existing models using additional reaction datasets with transfer learning techniques.
- This paper tries to apply three techniques (joint-training, self-training, and pre-training+fine-tuning) into a seq-to-seq model, Transformer. As reported in the paper, pre-training+fine-tuning achieves significant improvements over the baseline. The obtained results are comparable with SOTA methods that utilize different knowledge (e.g., atom-mapping or templates) instead of the augmented datasets.

---

Concern \#1: Limited contribution
- This paper just applies the simplest transfer methods to the existing seq-to-seq model. Although transfer learning for retrosynthesis is an interesting topic, I think the methodology contribution is very limited.
- Additionally, there is no insightful (experimental) analysis.

Concern \#2: Experiments are not enough
- The provided experiments are insufficient to demonstrate the effectiveness of transfer learning. For example, it is hard to guarantee that the used transfer learning techniques are useful for other retrosynthesis frameworks (e.g., GLN or G2Gs). I want to note that the transfer learning literature in other domains often experiments with various scenarios (e.g., different architectures, tasks, datasets) to demonstrate the effectiveness.
- Why the gap between pre-training+fine-tuning and joint-training is too large? I think the gap is somewhat surprising, but there is no explanation and analysis of this.
- Is the top-k accuracy only one merit of transfer learning? In vision tasks, pre-training can improve robustness [Hendrycks'19]. If other metrics (e.g., diversity [Schwaller'19]) are evaluated, the paper would be stronger.

[Hendrycks'19] Using Pre-Training Can Improve Model Robustness and Uncertainty, ICML 2019 \
[Schwaller'19] Evaluation Metrics for Single-Step Retrosynthetic Models, Second Workshop on Machine Learning and the Physical Sciences (NeurIPS 2019)

---

Conclusion: The topic is interesting, but contribution seems to be very limited. More extensive experiments/analyses and/or a novel transfer technique specialized in retrosynthesis should be presented.

---

### Official Review · AnonReviewer2 · 2020-10-25
**Unsurprising empirical results given similarity in datasets; unclear if conclusions generalize beyond this application/task**

**Rating:** 4
**Confidence:** 4

**Review:**

This work examines different datasets and strategies for pretraining a Transformer model to perform one-step retrosynthesis as a SMILES-to-SMILES translation task. Three approaches are evaluated: joint training, joint training on self-labeled data, and pretraining/fine-tuning.

The related work section and problem formulation setting are rather long, but provide a reasonable overview of approaches used to date. It may behoove the authors to move some of the results and discussion from the appendix to the main text and shorten the introduction in a revision/resubmission.

The target dataset selected in this work is the USPTO-50K set, which has become a standard benchmark; that is a reasonable choice. The two larger datasets used for augmentation are “USPTO-Full” and “USPTO-MIT”. All three of these datasets were taken from the patent literature (if that wasn’t obvious by the name). In particular, they should all be strict subsets of USPTO-Full if the same preprocessing steps are used.

My major concern is that the pre-training + fine-tuning approach might not be a reasonable approach to use with these datasets. The pretraining setting is merely training on a larger dataset for the same task of identical noise/quality. The breadth of transformations in USPTO-Full is larger, but it should contain a superset of reactions and reaction types compared to USPTO-50K. This is a relatively minor domain shift, so it’s not surprising that the pretraining approach works well. I appreciate that the authors have removed exact matches where the product SMILES are identical (hopefully after removing atom mapping and canonicalizing – please confirm), but there will be an abundance of very similar training examples even without exact matches.

I would certainly not consider this approach to achieve the new state-of-the-art, as is claimed in the abstract because it is merely using additional data for training, which other studies *intentionally* do not do to allow for a fair comparison.

The results in Figure 2 showing the tradeoffs between top-1 and top-20 test accuracy are interesting. This does suggest using the validation set to monitor for overfitting is worthwhile.

There is a claim that over 99% of top-1 predictions are reasonable and appropriate hypotheses. However, there is no way to evaluate this because—as far as I can tell—this data is not contained in the submission.

This work does not make any significant contributions from an algorithmic or theoretical perspective and requires additional clarifications and discussion to justify the significance of its empirical results. I do find aspects of the analysis novel, but those aspects might be more interesting to an application- or domain-focused venue.

Factual corrections:
-	The statement “any compound can be uniquely represented as a SMILES format string” is potentially confusing. Many distinct SMILES strings can describe the same compound; SMILES strings are not unique.
-	The USPTO-50K dataset was not curated by Lowe (2012). It is from https://doi.org/10.1021/ci5006614
-	The dataset released by Lowe contains over one million reaction examples; is the 877K number after filtering?


Minor corrections:
-	Figure 1 labels retrosynthesis as $p(X|Y)$ rather than $p(Y|X)$; the notation of $X$ and $Y$ is slightly inconsistent with the notation in 3.1 using $x$ and $y$.
-	Parameters in the likelihood function are referred to as both $\theta$ and $\theta_\mathcal{M}$
-	Optional parameters are referred to as both $\theta^*$ and $\theta_{\mathcal{M}^*}$
-	The notation for argmax expressions is a little unconventional

---

### Official Review · AnonReviewer3 · 2020-10-27

**Rating:** 4
**Confidence:** 5

**Review:**

### Summary of the paper
This paper proposes to improve retrosynthesis models with pre-training and self-training techniques. For pre-training, the model is trained on the USPTO reaction dataset and fine-tuned on USPTO-50K dataset. For self-training, the model is trained on artificial reaction instances generated by the model (i.e., back-translation). The pre-training based approach greatly improves the seq2seq retrosynthesis model and achieves comparable performance against graph based methods.

### Strength
1) The paper conducts extensive experiments for various training paradigms: joint training, self-training and pre-training. For pre-training, authors try different options of the pre-training dataset (USPTO, USPTO-MIT).
2) The self-training approach is an interesting application of back-translation.

### Weakness
1) Technical novelty is very weak. The pre-training approach is not novel. It is simply training on a larger training set, leveraging additional data. The self-training approach is not novel either -- it is a simple application of back-translation, which is well-studied in machine translation in NLP.
2) The performance is weak. From Table 2, we can see that current state-of-the-art are graph-to-graph based approaches. The best result of this paper is worse than the state-of-the-art (for 1,3,5-best accuracy), even though it is trained on additional data.
3) In principle, the pre-training method is not limited to seq2seq retrosynthesis models, but authors only apply pre-training for the seq2seq transformer architecture, which performs much worse than graph-to-graph methods.

### Overall evaluation and suggestions
I vote for rejection. Unfortunately, the weakness of the paper greatly outweighs the strength. I am afraid the technical novelty is too weak for ICLR. The results are unsatisfactory due to lower 1-best accuracy compared to prior methods. To improve the paper, authors should try to apply pre-training to graph-to-graph models, which may lead to new state-of-the-art results. Since your pre-training is just training on a larger dataset, it shouldn't be too hard to do.

### Post rebuttal
I would like to thank all the reviewers for valuable feedback. My review score stays the same.

---

### Official Review · AnonReviewer1 · 2020-10-28
**Interesting but premature investigation of data transfer in retrosynthesis**

**Rating:** 4
**Confidence:** 5

**Review:**

The paper uses finetuning for SEQ2SEQ translation models to improve performance for retrosynthesis prediction. In particular, transformer models are finetuned on a large dataset to improve performance on a smaller dataset.
Such an investigation of fine tuning approaches for retrosynthesis is interesting for the experts working in the area and therefore it is much appreciated that the authors did this study.

While promising, however, the work is still premature. Unfortunately, the performed experiments do not fully support the claims made, in particular, the investigation is neither "systematic and intensive". Furthermore, a SOTA baseline is not cited (MEGAN, see below), which means the approach reported here actually does not achieve SOTA in any task, in contrast to what the authors claim.

The underlying idea of the paper is somewhat incremental and lacks novel insight on the ML side. With additional experiments, the paper might be better suited for a good chemoinformatics journal where it would find an expert audience, however, for a leading ML conference, this is too little. Transfer learning for reaction prediction transformer models has already been explored by Pesciullesi et al. (2020) and Chen et al. (2019).

This reviewer is not convinced that the comparison provided in the paper are meaningful.
All of the models in the study, not just Transformer, could be improved by finetuning. In a "systematic and intensive" study, the authors should apply the fine tuning to the other models as well, e.g. GLN, RetroXpert, G2G, GraphRetro, and others, and not just to Transformers, where all of the hard work of implementation has been done already by someone else. Given that Transformer without finetuning is outperformed by all of these models, it is likely that finetuning these other models on the larger dataset will also outperform finetuned Transformer models. These key experiments are missing.

regarding the experimental setup used and data:
The USPTO50k dataset the authors use for validation is a random subset of the larger USPTO full dataset. It is therefore no surprise that a model (pre)trained on the full dataset performs better than a model trained only on a subset of that dataset.

Why do the authors not compare the performance on the full dataset then directly? In a realistic scenario, the user would train on the largest dataset available, and not on a small subset.

Missing references:

- Segler et al perform fine tuning of sequence based models on SMILES https://arxiv.org/abs/1701.01329 (2017)
- Winter et al perform pretraining using sequence to sequence models on SMILES https://doi.org/10.1039/C8SC04175J (2018)
- Sacha et al report the MEGAN model https://arxiv.org/abs/2006.15426 which performs better than the fine-tuned transformer presented here in in top10 and top20 accuracy (87.6, 91.6) vs (87.4, 89.6).

---

### Author Response · Authors · 2020-11-16
**Thank you all reviewers**

We thank all reviewers for their insightful comments and interesting proposals.
We are encouraged that some reviewers feel positive to the topic (transfer learning) itself.
We will improve the manuscript following your advice and look for other venues.

To reviewer 1
- We are not aware of the MEGAN paper. Thank you for pointing out missing important references.
- As you suggest, we will consider submitting an improved manuscript for chemoinfo journals.
- We would consider training graph2graph models with full dataset.

To reviewer 2
- Our main message is that the training on the larger data drastically (20% is simply surprising)  improves the retrosynthesis prediction... it means we should work on the USPTO-FULL dataset instead of 50K subset to compete in model developments.
- Try pertaining+fine-tune to graph2graphs is a reasonable proposal. Thank you.

To reviewer 3
- Your comments on the minor domain shift is new and intersting for us. Thank you.
- Thank you for factual correctness. These are very helpful for our future paper preparations.

To reviewer 4
- We appreciate your interesting proposal to quantify the effectiveness of transfers by other metrics. We will check the referred articles and think about new metrics we can validate.

---

### Decision · Program_Chairs · 2021-01-07
**Final Decision**

**Decision:**

Reject

**Comment:**

While the authors thought that the paper had some strong experimental comparisons, there were serious concerns with novelty and paper claims. For a stronger ML paper the authors would need to either: (a) design a new training methodology beyond pre-training that is better suited for leveraging multiple datasets for Retrosynthesis, (b) design a new model for Retrosynthesis that is better able to leverage mutliple datasets, (c) design new evaluation metrics to describe how well current methods perform in Retrosynthesis and/or metrics that describe how well methods can use data from different sources. That said, if the authors were interested to submit to non-ML venues then I agree with R2 that chemistry venues may be better suited to the paper in its current form.